# Deworming coverage and its determinants among 12–59 months old children in East Africa: A population-based study

**Bewuketu Terefe**[1]*, **Mahlet Moges Jembere**[2], **Nega Tezera Assimamaw**[3], **Bogale Chekole**[4]

1 Department of Community Health Nursing, School of Nursing, College of Medicine and Health Sciences, University of Gondar, Gondar, Ethiopia, 2 Department of Emergency and Critical Care Nursing, School of Nursing, College of Medicine and Health Sciences, University of Gondar, Gondar, Ethiopia, 3 Department of Pediatric and Child Health, School of Nursing, College of Medicine and Health Sciences, University of Gondar, Gondar, Ethiopia, 4 Department of Comprehensive Nursing, College of Medicine and Health Sciences, Wolkite University, Welkite, Southern Ethiopia

* woldeabwomariam@gmail.com

## Abstract

### Background

Intestinal parasitic infections are the world's largest public health issue, primarily in developing nations. The World Health Organization (WHO) recommends deworming as a preventative or therapeutic measure for all vulnerable people residing in endemic areas. Despite this issue, there is little data on the prevalence and associated factors of deworming drug use among children under five years of age in East Africa.

### Objective

This study aimed to evaluate the prevalence and contributing factors of deworming coverage among children under the age of five in East Africa using the most available national health survey data.

### Methods

Data from the Demographic and Health Survey, which included 103,865 weighted children between the ages of 12–59 months, were used in this investigation. Our outcome of interest was taking deworming medicine six months before the interview. A logistic regression model was then fitted. A cutoff P value of 0.2 was used in the binary logistic regression analysis. To identify significant variables, a 95% confidence interval and adjusted odds ratio (AOR) with a value < 0.05 were used.

### Results

The prevalence of deworming in East Africa was 54.13% (95% CI: 53.83%–54.43%). The maternal age group of 24–34 years, and from 35–49 years (AOR = 1.37, 95% CI, 1.32,1.42), and (AOR = 1.71, 95% CI, 1.62,1.79), employed women (AOR = 1.62, 95% CI,

**Data Availability Statement:** All this study files are available from the DHS MEASURES database at https://www.dhsprogram.com/.

**Funding:** The author(s) received no specific funding for this work.

**Competing interests:** The authors have declared that no competing interests exist.

1.58,1.67), being from rural(AOR = 1.11,95% CI,1.07,1.15), unmarried mothers (AOR = 1.12,95% CI,1.09,1.15), mothers from poorer, middle, richer, and richest households (AOR = 1.16,95% CI, 1.12,1.21), (AOR = 1.23, 95% CI, 1.18,1.28), (AOR = 1.22,95% CI, 1.16,1.27), and (AOR = 1.27, 95% CI, 1.21,1.34) having at least one antenatal care follow up(AOR = 2.90, 95% CI, 2.63,3.16), health facility delivery(AOR = 1.69, 95% CI,1.64,1.75), mass media exposure AOR = 1.32, 955 CI, 1.29,1.36), having of 3–5 children (AOR = 0.89, 95% CI, 0.86,0.93), more than five children (AOR = 0.79, 95% CI, 0.73,0.86), and parity of 2nd or 3rd birth order (AOR = 1.05, 95% CI, 1.01,1.09) as compared to primi mothers were associated with the deworming among under five children in east Africa respectively.

## Conclusion

The under-five population in East Africa had a lower prevalence of deworming medication per the most recent DHS findings. Promoting mother and child health services (antenatal care, institutional delivery, family planning), as well as women's empowerment, should be prioritized.

## Introduction

Soil-transmitted helminth (STH) infections are the most frequently neglected tropical diseases (NTDs) worldwide, impacting approximately 1.5 billion people or 24% of the global population [1, 2]. STH infections are widespread in tropical and subtropical regions of the world, with the highest prevalence in sub-Saharan Africa (SSA), the Americas, China, and East Asia. However, bears the highest load [2, 3]. More than 10.5 million new cases are recorded each year, with the most prevalent intestinal parasites being Ascaris lumbricoides, hookworms, Trichuris trichiura, Giardia lamblia, Entameba histolytica, and Schistosoma's [4].

STH infections spread from person to person through soil contaminated with human feces [3]. They frequently affect children living in poverty in an unsanitary setting [5]. According to the World Health Organization (WHO), approximately 270 million preschool-aged children (pre-SAC) live in areas where these parasites are disseminated [2, 5, 6].

According to a WHO assessment, approximately 270 million preschools and 600 million school-aged children live in places where parasites are widely transmitted and require treatment and prevention [7]. Between 2004 and 2017, the average global deworming coverage of preschool children was projected to be 36% in 50 STH-endemic countries. Subnational coverage ranged from 0.5% to 87.5%, with within-country variation in coverage being greater than between-country variation [8]. In endemic nations, nearly 300 million preschool and school-age children were dewormed in 2009, accounting for 35% of at-risk children. The global goal is to reach at least 75% of these vulnerable youngsters [9].

In places where helminth infection is endemic, it is suggested that all preschool-aged children be treated with deworming medications at 6-month intervals [10]. The strategic plan called for STH to be eliminated as a public health issue for children by 2020 [6]. As a public health intervention, annual or biannual preventive chemotherapy (deworming) with a single dose of albendazole or mebendazole is recommended for all young children 12–23 months of age, preschool children 2–5 years of age, and school-age children 6–12 years of age living in areas where the baseline prevalence of any soil-transmitted infection among children is 20% or higher [11].

East African countries have launched targeted deworming programs to prevent intestinal parasite infections in preschool-age children to reduce morbidity and death [6, 7, 12–14]. Despite the availability of selective deworming medications for children under the age of five years and the promotion of health education through health extension workers, intestinal parasitic infections remain the leading cause of morbidity and mortality. This hinders efforts to create sustainable growth. Little is known about deworming and its predictors in children under five years of age. The WHO recommends that all children receive deworming after the age of one year [14]. Using a recent national health survey, this study aimed to examine deworming coverage and possible associated factors in East African children aged 12 to 59 months.

## Methods

### The study design, setting, and time frame

The data came from the most recent standard Demographic and Health Survey (DHS) dataset of East African countries over 10 years (2011/12–2022). We used a standardized dataset [15] to obtain all parameters and a large sample size that is representative of the population source. The DHS collects cross-nationally comparable data. The surveys are population-based and nationally representative of each country, with large sample sizes [15]. Eastern Africa is composed of 14 countries located in the Great Lakes region, Horn of Africa, and Indian Ocean islands. These countries face similar economic, social, and environmental issues and are concerned about not meeting all of the Millennium Development Goals targets [16]. East Africa is a region of the African continent that falls within the Saharan desert in Horn and eastern Africa. According to United Nations estimates, they cover an area of 6,667,493 Km2 (2,574,332 square miles) and house 6.03% of the world's population, with a total population of 486,766,759.

### Source and study population

Our source population included all children under five years before the survey period in 12 East African countries (Table 1), while our study population included children aged 12–59 months in the designated Enumeration Areas (EAs) or primary sample units of the survey clusters. In each country, the mother or caregiver was questioned for the survey, and mothers who had more than one child within the previous two years were asked questions regarding

**Table 1. Countries, sample size, and survey year of Demographic and Health Surveys included in the analysis for 12 East African countries.**

| Country | Survey year | Sample size(weighted) | Frequency(weighted) |
|---|---|---|---|
| Burundi | 2016/17 | 10,101 | 9.72 |
| Ethiopia | 2016 | 8,092 | 7.79 |
| Kenya | 2022 | 13,218 | 12.73 |
| Comoros | 2012 | 2,914 | 2.81 |
| Madagascar | 2021 | 8,999 | 8.66 |
| Malawi | 2016/17 | 13,074 | 12.59 |
| Mozambique | 2011 | 10,375 | 9.99 |
| Rwanda | 2019/20 | 6,361 | 6.12 |
| Tanzania | 2015/16 | 7,443 | 7.17 |
| Uganda | 2016 | 11,236 | 10.82 |
| Zambia | 2018 | 7,212 | 6.94 |
| Zimbabwe | 2015 | 4,840 | 4.66 |

the most recent child [15]. Furthermore, from the dataset of the included countries, children aged 12–59 months who were not assessed for deworming medicine based on the DHS recommendation and had a missing value of the outcome variable were omitted. Similarly, children who had just died were not included in the study, based on the DHS recode manual. Nonetheless, the study includes missing values and "don't know" responses regarding the child's use of deworming medication in the two years before to the interview, but these responses are not regarded as dewormed [15].

## Sample size determination and sampling method

Approximately 12 of the 13 East African countries have Demographic and Health Survey Reports. All surveys conducted in listed countries used the most current conventional census frames. DHS samples are often stratified by administrative geographic region and urban/rural area within each region. Enumeration areas (EAs) were chosen with a probability proportional to the size of each stratum in the first round of sampling. In the second stage, the systematic sampling approach selects a predefined number of households in the specified EAs. Following the listing of the households, a fixed number of households were chosen in the designated cluster using equal-probability systematic sampling [15].

## Data sources

The DHS databases for children's records and Child's records (KR) were utilized. Before using the DHS dataset, weighted values were used to restore the representativeness of the sample data. This is because the total chance of selecting each household is not constant. The DHS standards established four sampling weighting methods, one of which was used for women (v005). Individual sample weights were calculated by dividing (v005) by 1,000,000 and then used to estimate the number of cases [17]. Finally, this study included a total weighted sample of 103,865 children aged 12–59 months from all 12 nations.

## Data collection tools, and quality control

The DHS uses interviewer-administered questionnaires to gather data through several questions. The DHS guidelines define the missing values in the outcome variables in a straightforward manner. However, as complete case analysis is a preferable way to manage missing data in a cross-sectional study, variables with a missing value greater than 5% in the explanatory variables were removed from further analysis. To guarantee quality, data extraction was performed by public health specialists with prior experience working with DHS data.

## Data processing and analysis

The standard DHS dataset was downloaded in the STATA format before being cleaned, integrated, transformed, and appended to provide useful variables for the analysis. To define variables in the study using statistical measurements, Microsoft Excel 2019 and STATA version 17 software were used to obtain both descriptive and analytical statistics [18]. In the bivariable analysis, variables having a p-value of equal to or less than 0.2 were considered for the multivariable analysis. Because the data could be hierarchical, we examined it for the assumption of multilevel model analysis using the intraclass correlation (ICC) coefficient, but it was <5%, which did not meet the minimal criterion to conduct it. As a result, classical logistic regression was preferable. The best-fitted model's Adjusted Odds Ratio (AOR) with 95% confidence interval (CI) was provided in the multivariable logistic model to identify the associated factors of deworming medication. Descriptive studies, such as frequency counts and proportions for

categorical data, were used to summarize the descriptive data. Bivariate logistic regression was employed to select candidate variables for multiple logistic regression. Using the variance inflation factor, a logistic regression was fitted to examine multicollinearity among the independent variables. The Hosmer-Lemeshow test was also employed to evaluate the overall fitness of the final regression model. Statistical significance of the final model was set at p < 0.05.

## Variables of the study

**The outcome variable.**   The outcome variable of this study was the number of living children 12–59 months who received deworming medication in the six months preceding the interview. The outcome variable was then categorized as "Yes" with a value of 1, if the child received any deworming medication, if not it will be classified as "No" with a value of 0. This classification was made according to the guidelines in the DHS statistics book [15].

**Independent variables.**   Independent variables: Various maternal- and child-related factors were included. This included maternal age, educational status, types of places of residence, marital status, household wealth index, current employment status, mass media exposure, ANC follow-up, place of delivery, PNC checkup, number of health visits, and total children born, under-five children, contraceptive utilization, age of the child, sex of the child, size at birth, twin status, birth order, sex of the household head and countries were included.

**Operational definitions for the independent variables.**   Independent variables were classified after reviewing similar literatures as follows (Table 2).

**Ethical considerations and data access.**   The study was conducted after obtaining a permission letter from www.dhsprogram.com on an online request to access East African DHS data after reviewing the submitted brief descriptions of the survey to the DHS program. The datasets were treated at the highest confidence level. This study was based on secondary data from the East African DHS. Issues related to informed consent, confidentiality, anonymity, and privacy of the study participants were already addressed ethically by the DHS office. All the above ethical related issues are described in the permission letter. We did not manipulate

**Table 2. Operational classification of independent variables (Table 1).**

| | |
|---|---|
| Birth weight | The size of the child at birth was classified in the DHS data set as very small, smaller than average, average, larger than average, and very large depending on the mother's perception of the size of the child at birth. We reclassified as small-size babies, average-size babies, and large-size babies, falling into each of these categories |
| Total children ever born | Total number of ever born children was given in number in the DHS data set, and we have recategorized it 0–2, 3–5, and more than five children in the given household |
| ANC follow ups | Antenatal care follow-up was also expressed in number in the DHS data set, and we have recategorized it as Yes, if the mother has at least one ANC visit, unless No ANC follow. |
| Number of under five children | Under-five children in the given family have been put numerically in the DHS data set and we have re-arranged it as No, 1 or 2, and more than two under-five children, |
| Birth order | Order of birth was given numerically in the original data set, and as usual, we have reorganized it as first, from 2–3, 4–5, and > = 6 birth orders |
| Child age in months | The age of the child was given numerically in months, and we have used it as 12–23, and 24-35-, and 36-59-months old age. |
| Maternal age | Maternal age was classified as 15–24, 25–34 and 35–49 years old |
| Maternal education | Maternal education was classified as no formal education, primary, and secondary and high educational level |
| Mass media exposure | This was given as watching to television, listening to radio, and reading to newspapers/book. To get the mass media exposure status the above three means were added up, and those mothers who have an exposure at least one means were taken as yes, unless no |
| Wealth index | Household wealth index was taken as it is in the data set (poorest, poorer, middle, richer, and richest) |

or apply microdata other than those used in this study. Since, no patient or public involvement in this study, only permission letter from the DHS MEAURE was enough, and no need of ethical clearance letter.

## Results

### Sociodemographic characteristics of the study participant

In this study, a weighted total of 103,865 children whose age was–12–59 months old were enrolled in East African countries. About nearly half 50,124 (48.26%) of the study women were found from the age group of 25–34 years of reproductive age. Regarding marital status majority of mothers, 71,278 (68.63%) of the participants were married. Regarding place of residence types 80,245 (77.26%), educational status 51,352 (49.44%), wealth index 24,318 (23.41%), place of delivery 80,688 (77.69%), and ANC follow-up, 99,836 (96.12%) of the mothers were from rural areas, primary educational status, poorest household's wealth index, institutional delivery, and had at least one ANC follow-up during their pregnancies. Similarly, about 55,456 (53.39%) women had at least one mass media exposure (either listening to the radio, watching television, or reading magazines/newspapers); however, 65,677 (63.23%) and 38,380(64.87%) women were employed and did not undergo postnatal checkups, respectively. Furthermore, only 55,272 (53.22%) and 80,049 (77.07%) participants did not use any type of contraceptive method and visited health facilities once in the past 12 months, respectively (Table 1). In addition, regarding child-related characteristics, the majority of mothers 43,603 (41.38%) had to 3–5 children, and almost all of them were single 100,982 (97.22%). Regarding the order of birth, approximately 37,947 (36.53%) were 2nd or 3rd birth order, with an average weight of 48, 001 (51.09%), and about 84, 775 (81.62%) mothers had children in their house. Finally, almost half of the children 52,282(50.32) were male (Table 3).

### Factors associated with deworming medication among 12–59 months old children in East Africa

The odds of being exposed to deworming medication among children increased by 37%, and 71% (AOR = 1.37, 95% CI, 1.32,1.42), and (AOR = 1.71, 95% CI, 1.62,1.79) more among women whose age is from 24–34 years, and from 35–49 years old, compared to women whose age found from a group of 15–24 years old respectively. Deworming medication reception was increased by 39% (AOR = 1.39, 95% CI, 1.34,1.44), and 41% (AOR = 1.41, 95% CI, 1.35,1.47) more among mothers who had completed their primary and secondary/higher educational attainment as compared to uneducated mothers, respectively. Mothers who were employed and came from rural areas showed 62% and 11% increased chances of providing deworming medication to their children (AOR = 1.62, 95% CI, 1.58,1.67) and (AOR = 1.11,95% CI,1.07,1.15) more times than unemployed and urban residential mothers, respectively. Regarding marital status, unmarried mothers showed a higher likelihood (AOR = 1.12,95% CI,1.09,1.15) of giving deworming medication to their children than married women. Regarding the household wealth index, mothers who came from poorer, middle, richer, and richest households had a higher likelihood of providing deworming medication to their children than the poorest household wealth index mothers by the odds ratios of (AOR = 1.16,95% CI, 1.12,1.21), (AOR = 1.23, 95% CI, 1.18,1.28), (AOR = 1.22,95% CI, 1.16,1.27), and (AOR = 1.27, 95% CI, 1.21,1.34) more times, respectively. Women who had at least one ANC visit and had given their children at health facilities showed (AOR = 2.90, 95% CI, 2.63,3.16), and (AOR = 1.32, 955 CI, 1.29,1.36) more times to medicate their children for deworming medication than mothers who had no ANC visits and had given their birth at home, respectively.

**Table 3. Maternal and child related sociodemographic characteristics of respondent's utilization of deworming medication consumption among 12–59 months old children in East Africa (weighted n = 103,865): Based one the recent East African countries DHS data.**

| Variables | Categories | Frequency | Percentage |
|---|---|---|---|
| Age in years | 15–24 | 28,421 | 27.36 |
| | 25–34 | 50,124 | 48.26 |
| | 35–49 | 25,320 | 24.38 |
| Residence | Urban | 23,621 | 22.74 |
| | Rural | 80,245 | 77.26 |
| Mothers' Educational status | No education | 24,856 | 23.93 |
| | Primary | 51,352 | 49.44 |
| | Secondary and higher | 27,658 | 26.63 |
| Mothers employed | No | 38,188 | 36.77 |
| | Yes | 65,677 | 63.23 |
| Wealth index | Poorest | 24,318 | 23.41 |
| | Poorer | 21,590 | 20.79 |
| | Middle | 20,062 | 19.32 |
| | Richer | 19,715 | 18.98 |
| | Richest | 18,181 | 17.50 |
| Mass media exposure | No | 55,456 | 53.39 |
| | Yes | 48,409 | 46.61 |
| ANC follow-ups | No | 4,021 | 3.88 |
| | Yes | 99,836 | 96.12 |
| Place of delivery | Home | 23,177 | 22.31 |
| | Health facility | 80,688 | 77.69 |
| Total number of children ever born | 0–2 | 38,348 | 36.92 |
| | 3–5 | 43,603 | 41.98 |
| | 6 & above | 21,914 | 21.10 |
| Contraceptive method types | No methods | 55,272 | 53.22 |
| | Traditional | 3,203 | 3.08 |
| | Modern | 45,390 | 43.70 |
| Number of health visits in the past 12 months | Once | 80,049 | 77.07 |
| | More than one | 23,816 | 22.93 |
| Marital status | Unmarried | 32,587 | 31.37 |
| | Married | 71,278 | 68.63 |
| Size of the child(n = 93,950) | Small | 15,389 | 16.38 |
| | Average | 48,001 | 51.09 |
| | Large | 30,560 | 32.53 |
| Sex of the child | Male | 52,282 | 50.34 |
| | Female | 51,583 | 49.66 |
| PNC checkup (n = 59,167) | No | 38,380 | 64.87 |
| | Yes | 20,788 | 35.13 |
| Child age in months(n = 90,575) | 12–23 | 23,059 | 25.46 |
| | 24–35 | 22,278 | 24.60 |
| | 36–59 | 45,238 | 49.95 |
| Under five children number | No | 3,290 | 3.17 |
| | 1–2 | 84,775 | 81.62 |
| | >02 | 15,800 | 15.21 |

(*Continued*)

**Table 3.** (Continued)

| Variables | Categories | Frequency | Percentage |
|---|---|---|---|
| Birth order | First | 24,899 | 23.97 |
| | 2nd or 3rd | 37,947 | 36.53 |
| | 4th or 5th | 22,654 | 21.81 |
| | Above 5th | 18,366 | 17.68 |
| Twin status | Single | 100,982 | 97.22 |
| | Multiple | 2,883 | 2.78 |
| Sex of the household head | Male | 79,172 | 76.23 |
| | Female | 24,693 | 23.77 |

Mothers who were born from 3–5 to, and more than five children showed lower odds of (AOR = 0.89, 95% CI, 0.86,0.93), and (AOR = 0.79, 95% CI, 0.73,0.86) compared to mothers who had given fewer than three children. However, 2nd or 3rd birth order children showed higher odds of being exposed to deworming medication than first-birth-order children (increases by 5%) (Table 4).

## Discussion

This study aimed to determine parameters related to the use of deworming medication as chemoprophylaxis among under-five children in East African countries. Based on this, the pooled estimate of deworming medicine utilization among under-five-year-old children in East African nations was 54.13% (95% CI: 53.83%, 54.43%), with Rwanda accounting for 88.79% and Ethiopia accounting for 13.65%. Our findings are higher than those of schistosomiasis and soil-transmitted helminthiases progress report, 2020, which has 9.21% coverage among 13 African countries [19], and studies conducted in 39 countries UNICEF offices (49.1%) [20], reports from SSA 45.03% [21], and global deworming programs that aim to reach 75% pre-SAC by 2020 [6, 20]. In contrast, this figure is lower than the national deworming coverage of Burundi, Myanmar, and the Philippines, where 80.1%, 93.6%, and 75.7% of children, respectively, received deworming supplementation [22]. According to a study conducted in Zambia, the health campion's deworming coverage rate in 2012 was 93.4%, which is greater than this finding [23].

These variances could be attributed to differences in sociocultural factors among study participants, data collection time frames, and differences in awareness and familiarity with the need for deworming to prevent STH infections in children under the age of five [24, 25]. This information was gathered using standardized, identical questionnaires. Even if there was no statistically significant difference, the higher prevalence of deworming in non-endemic nations may be attributable to increased health-seeking behavior and deworming program adjustments, eventually leading to STH infection control. These additional discrepancies could be due to the difference in sample sizes between the two groups.

The age of women and deworming drug use were positively related. More precisely, we discovered that older women had higher rates of deworming medicine use than younger women. Higher levels of educational and professional achievement, as well as greater family income that increases with age, may all be contributing factors [26, 27]. Older women are typically proactive and focus on disease prevention [26, 28], which may be related to their prior experience. A study from SSA reported a similar association with women's age [26, 29]. This issue can be partially revealed by raising awareness among young mothers.

According to this study, children with mothers who attended primary and secondary/higher education were 41% and 39% more likely, respectively, to receive deworming

**Table 4. Multiple logistic regression analysis results on determinants deworming medication provision among 12–59 months age children in East African (weighted n = 103,865, and unweighted n = 104,164): Based one the recent East African countries DHS data.**

| Deworming medication is given | No, n (%) | Yes, n (%) | COR 95% | AOR (95% CI) |
|---|---|---|---|---|
| **Variables** | | | | |
| **Maternal age** | | | | |
| 15–24 | 14,086(49.56) | 14,335(50.44) | 1 | 1 |
| 25–34 | 22,380(44.65) | 27,744(55.35) | 1.23(1.19,1.27) | **1.37(1.32,1.42)** |
| 35–49 | 11,178(44.15) | 14,142(55.85) | 1.27(1.23,1.32) | **1.71(1.62,1.79)** |
| **Maternal education** | | | | |
| Not educated | 14,526(58.44) | 10,330(41.46) | 1 | 1 |
| Primary | 22,674(44.15) | 28,678(55.85) | 1.81(1.76,1.87) | **1.39(1.34,1.44)** |
| Secondary & higher | 10,444(37.76) | 17,214(62.24) | 2.19(2.12,2.27) | **1.41(1.35,1.47)** |
| **Mother is employed** | | | | |
| No | 20,838(54.57) | 17,350(45.43) | 1 | 1 |
| Yes | 26,806(40.81) | 38,871(59.190 | 1.77(1.73,1.82) | **1.62(1.58,1.67)** |
| **Residence** | | | | |
| Urban | 9,366(39.65) | 14,255(60.35) | 1 | 1 |
| rural | 38,278(47.70) | 41,967(52.30) | 0.79(0.77,0.81) | **1.11(1.07,1.15)** |
| **Marital status** | | | | |
| Married | 33,657(47.22) | 37,621(52.78) | 1 | 1 |
| Unmarried | 13,986(42.92) | 18,601(57.08) | 1.19(1.16.1.22) | **1.12(1.09,1.15)** |
| **Wealth index** | | | | |
| Poorest | 13,180(54.20) | 11,138(45.80) | 1 | 1 |
| Poorer | 10,497(48.62) | 11,093(51.38) | 1.41(1.35,1.46) | **1.16(1.12,1.21)** |
| Middle | 9,018(44.95) | 11,044(55.05) | 1.61(1.55,1.67) | **1.23(1.18,1.28)** |
| Richer | 8,437(42.80) | 11,277(57.20) | 1.71(1.65,1.78) | **1.22(1.16,1.27)** |
| Richest | 6,511(35.81) | 11,670(64.19) | 2.06(1.98,2.14) | **1.27(1.21,1.34)** |
| **ANC follow-ups** | | | | |
| No | 3,246(80.57) | 783(19.43) | 1 | 1 |
| Yes | 44,398(44.47) | 55,439(55.53) | 4.87(4.49,5.28) | **2.90(2.67,3.16)** |
| **Place delivery** | | | | |
| Home | 14,404(62.15) | 8,773(37.85) | 1 | 1 |
| Health facility | 33,239(41.19) | 47,449(58.81) | 2.19(2.14,2.27) | **1.69(1.64,1.75)** |
| **Mass media exposure** | | | | |
| No | 29,062(52.40) | 26,394 (47.60) | 1 | 1 |
| Yes | 18,582(38.39) | 29,827(61.61) | 1.72(1.68,1.76) | **1.32(1.29,1.36)** |
| **Total children born** | | | | |
| >3 | 16,577(43.23) | 21,771(56.77) | 1 | 1 |
| 3–5 | 19,896(45.63) | 23,707(54.37) | 0.95(0.92,0.97) | **0.89(0.86,0.93)** |
| = >6 | 11,171(50.97) | 10,744(49.03) | 0.77(0.75,0.79) | **0.79(0.73,0.86)** |
| **Birth order** | | | | |
| 1st | 11,073(44.47) | 13,826(55.53) | 1 | 1 |
| 2nd or 3rd | 16,508(43.50) | 21,439(56.50) | 1.07(1.04,1.11) | **1.05(1.01,1.09)** |
| 4th or 5th | 10,694(47.21) | 11,960(52.79) | 0.95(0.92,0.99) | 0.97(0.91,1.03) |
| > 5th | 9,369(51.01) | 8,997(48.99) | 0.83(0.79,0.86) | 0.95(0.87,1.05) |

supplements than children whose mothers did not attend formal education. This result is consistent with research conducted in Ghana [30], and SSA [21]. This might be because when mothers' educational levels rise, their understanding of and skills in providing care for their children will advance [31, 32].

According to this study, children whose mothers were working at the time of the survey had a 62% higher chance of becoming dewormed than children whose mothers were not working. This is corroborated by studies conducted in Ghana [30], SSA [21], and Ethiopia [24], which found that working mothers were more likely to deworm their children than their counterparts who were not working. Therefore, employed individuals may be exposed to the value of supplements and availability of prescription drugs. They may have been exposed to a deworming culture through their social interactions and information exchanges with their coworkers [33, 34]. Similar to other studies, children who lived in wealthier households were more likely to take deworming medication than children who lived in poorer homes. The fact that affluent women are more likely to use deworming medications than poorer women supports this [21, 35]. This could be related to the use of free-camping supplements or oral deworming medicine. In terms of marital status, single mothers demonstrated a higher propensity to administer deworming drugs to their children than married women did. Although the author was unable to locate any older literature to support the claim, it is still possible that it is justified. Compared with single women, married women may have more duties, more children, and less control. Given their increased ability to make decisions, single mothers might teach their children to prioritize their health.

Compared with children who lived in urban areas, children in rural areas were more likely to take deworming medication. However, a study in Ghana and SSA [21, 30] found the opposite result. This may be because deworming programs and health services are readily available and accessible in urban areas. Socioeconomic disparities between rural and urban areas may have contributed to this discrepancy. However, we authors thought that beyond these uncovered facts between urban and rural areas, the reason why rural children had received deworming than urban children may be due to their frequent exposure these mothers will take their children to medical facilities or might provide deworming medication to their children to stop the illness, then they report that they have admitted deworming to their children.

Compared to women who had no ANC visits and had given birth at home, we discovered that women who had ANC visits and had given birth in medical facilities had higher odds of using deworming medicine. As more information about the advantages of deworming and good pregnancy outcomes is learned through counseling during repeated ANC visits, institutional delivery, and other maternal and child-related health services [29, 36, 37], ANC visits and institutional delivery are good opportunities to receive deworming medication. To strengthen the use of ANC and institutional delivery services in the region, as recently recommended by the WHO, governments should reduce or eliminate barriers such as long travel distances to health facilities that may be worsened by challenging geographic or road conditions [36]. The health sector should also act as a coordinator and partner with other sectors to increase its uptake.

The likelihood of deworming children under the age of five decreases as household size or total number of children increases. This is comparable to the SSA [21], and Ethiopian studies [24]. This may be because large families require more time and care. Consequently, it could be challenging to find adequate childcare for children [24].

This study also revealed that children were 32% more likely to receive deworming supplements if their mothers were exposed to the media. According to a study conducted in SSA [21], Ethiopia [33], and Nigeria [38], maternal media exposure increases maternal understanding of health care. Utilizing mass media exposure effectively will have a significant impact on maternal and child health because it has strong potential to change mothers' knowledge, attitudes, behaviors, and interests in their own and their children's health. Another Indian study that demonstrated that media exposure improves the use of maternal health services backed it up as well [39].

The utilization of a sizable sample of data that was individually weighted and reflective of the levels in each nation, as well as the region of East Africa, was the study's main strength. However, because the data were secondary and cross-sectionally gathered, they were subject to recall errors and social desirability biases. Another drawback is that not all populations in each country or sub-nation in East Africa would have the same greater frequency of soil-transmitted helminths, making them populations that might not require routine deworming. However, we were unable to obtain a national report on the prevalence of soil-transmitted helminths in each country or subnation of East Africa during the DHS year. Further robust primary studies with spatial distribution may provide a good opportunity to address this problem.

## Conclusions

According to the most recent DHS findings, East Africa's under-five population has a lower recommended prevalence of deworming medicine. The use of deworming medication among children under five years of age in East Africa was significantly influenced by several variables, including the age and educational status of women, parity, birth order, media exposure, ANC follow-ups, place of delivery, marital status, household wealth status, and residence. to achieve and sustain the first of the six worldwide targets for soil-transmitted helminthiases set for 2030, thereby eliminating STH morbidity among children. Therefore, international organizations such as the WHO, UNICEF, intergovernmental organizations, governments of each country, and other interested bodies should work as an integrated approach with other stakeholders to strengthen women's education, household, and media exposure to increase the utilization of deworming medication among children in East Africa. Priority should be given to promoting maternal and child health services (ANC, institutional delivery, family planning) as well as women's empowerment.

## Acknowledgments

We would like to acknowledge the DHS program for providing permission for this study following research ethics.

## Author Contributions

**Conceptualization:** Bewuketu Terefe.

**Data curation:** Bewuketu Terefe, Nega Tezera Assimamaw, Bogale Chekole.

**Formal analysis:** Bewuketu Terefe, Mahlet Moges Jembere.

**Funding acquisition:** Mahlet Moges Jembere, Nega Tezera Assimamaw, Bogale Chekole.

**Methodology:** Bewuketu Terefe, Bogale Chekole.

**Project administration:** Nega Tezera Assimamaw.

**Resources:** Bewuketu Terefe, Nega Tezera Assimamaw, Bogale Chekole.

**Software:** Bewuketu Terefe, Mahlet Moges Jembere, Nega Tezera Assimamaw.

**Supervision:** Mahlet Moges Jembere, Bogale Chekole.

**Validation:** Bewuketu Terefe, Mahlet Moges Jembere, Nega Tezera Assimamaw, Bogale Chekole.

**Visualization:** Bogale Chekole.

**Writing – original draft:** Bewuketu Terefe, Bogale Chekole.

**Writing – review & editing:** Mahlet Moges Jembere, Nega Tezera Assimamaw, Bogale Chekole.

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
