## [Decision Letter · Decision Letter 0]

12 Dec 2023

PONE-D-23-32930Deworming coverage and determinates among 12-59 months old children in East Africa: a population-based study using recent East African national survey data of 2011-2022PLOS ONE

Dear Dr. Terefe,

Thank you for submitting your manuscript to PLOS ONE. After careful consideration, we feel that it has merit but does not fully meet PLOS ONE’s publication criteria as it currently stands. Therefore, we invite you to submit a revised version of the manuscript that addresses the points raised during the review process.

We look forward to receiving your revised manuscript.

Kind regards,

Ayele Mamo Abebe, MSc in pediatric and child health nursing

Academic Editor

PLOS ONE

Journal Requirements:

Reviewers' comments:

Reviewer's Responses to Questions

**Comments to the Author**

1. Is the manuscript technically sound, and do the data support the conclusions?

Reviewer #1: Yes

Reviewer #2: Yes

2. Has the statistical analysis been performed appropriately and rigorously? 

Reviewer #1: Yes

Reviewer #2: No

3. Have the authors made all data underlying the findings in their manuscript fully available?

Reviewer #1: Yes

Reviewer #2: Yes

4. Is the manuscript presented in an intelligible fashion and written in standard English?

Reviewer #1: Yes

Reviewer #2: No

5. Review Comments to the Author

Reviewer #1: Review comments on Manuscript Number: PONE-D-23-32930 “Deworming coverage and determinates among 12-59 months old children in East Africa: a population-based study using recent East African national survey data of 2011-2022"

Overall, the idea of research is very interesting to be studied nowadays and paper is coherently developed. However, there are some comments and suggestions.

General comments the authors need to review the manuscript grammar and editing.

Title

- You may consider change the title to [Deworming coverage and determinates among 12-59 months old children in East Africa: a population-based study]

Abstract

- Change the subtitle [introduction] to [background and objectives] or [background and aim]. And try to make it more concise.

- It is recommended to write keywords in alphabetical order.

- You may consider changing the keywords to more representative words [Deworming, Soil-transmitted helminth infections, tropical diseases…]

- The results section is relatively long with extensive statistical results that are clear enough in the results and tables. Try to make it more concise.

Introduction

- Well structured

Subjects and methods

- The inclusion/exclusion criteria are not clear. Are there any factors may affect the results.

Statistical analysis

- Well structured

Discussion

- Well structured

Reviewer #2: Intestinal parasite infections are public health concerns in developing countries which makes the research area is relevant. However, I have some questions and comments on the research work.

Abstract section

1. Avoid using passive sentences. This also applies for the rest of the manuscript sections.

2. on introduction section line 24, please replace "risk factors" with "associated factors".

3. result section should be written in a manner that attracts the readers attention. e.g. line 41 "having of 3-5, and more children" looks like an incomplete sentence. furthermore, details should be minimized as much as possible to capture the main finding of the study. I highly recommend to rewrite this section

4. avoid abbreviations from the abstract section if possible. abbreviations like "ANC" should be written in full on its first appearance. line 39. This was also noted on the rest of the manuscript.

Introduction section

It is well organized and provides all the necessary background information.

1. line 103 "literature neglects children under the age of one year, and little is known in East Africa". The rationale for starting deworming among those aged 1 year and above is due to the low likelihood of acquiring the infection even in an endemic setting. The sentence creates a wrong image please remove it.

Method section

1. source and study population: It is not clear how your source and study population are completely different. It needs further elaboration.

2. Table 1: the sample size for Ethiopia looks like a typing error. please correct it.

3. Operational definition used by the survey should be available for some of the variables included. e.g traditional contraceptive method, size of the child, wealth index class etc.

result section

1. line 206-207. it is either a 37% increased odds or 1.37 times higher odds not 37% times. In addition, please provide the result next to the category or the variable in consideration to make it more readable.

2 line 209-210. same as above as well as incomplete sentences and missing figures

3. line 216. "a higher likelihood times"?

4. line 220, 223 missing values

5. line 224 "compared to with mothers".

I wrote those to point out as an example. It is full of grammatical errors and missing figures making it unreadable. please rewrite the result section properly and in an organized manner.

The research idea raised was good but at this point it is difficult to give a review beyond the result section.

6. PLOS authors have the option to publish the peer review history of their article (what does this mean?). If published, this will include your full peer review and any attached files.

Reviewer #1: No

Reviewer #2: No

---

## [Author Response · Author response to Decision Letter 0]

14 Dec 2023

Response to reviewers 

Dear editor, thank you very much for considering our article in your esteemed journal. Dear reviewers, we would like to thank you for your scholarly contribution to our manuscript. Dear reviewers, we have considered all your comments without jumping anyone of them, however, most of our reply are appended in the revised manuscript. Hence, we kindly request to consider it 

1. Is the manuscript technically sound, and do the data support the conclusions?

Reviewer #1: Yes

Reviewer #2: Yes

Dear reviewers, thank you very much for your cross validation 

2. Has the statistical analysis been performed appropriately and rigorously?

Reviewer #1: Yes

Reviewer #2: No

Dear reviewer, we have performed the statistical analysis using the appropriate model based on the data nature. We have also now improved the manuscript. 

3. Have the authors made all data underlying the findings in their manuscript fully available?

Reviewer #1: Yes

Reviewer #2: Yes

Dear reviewers, thank you 

4. Is the manuscript presented in an intelligible fashion and written in standard English?

Reviewer #1: Yes

Reviewer #2: No

Dear reviewer, we have tried our best to correct anything which has gone wrong including typos

5. Review Comments to the Author

Reviewer #1: Review comments on Manuscript Number: PONE-D-23-32930 “Deworming coverage and determinates among 12-59 months old children in East Africa: a population-based study using recent East African national survey data of 2011-2022"

Overall, the idea of research is very interesting to be studied nowadays and paper is coherently developed. However, there are some comments and suggestions.

General comments the authors need to review the manuscript grammar and editing.

Title

- You may consider change the title to [Deworming coverage and determinates among 12-59 months old children in East Africa: a population-based study]

Dear reviewer, thank you very much for your illustration. Partly based on your recommendation we have adjusted the title this way “Deworming coverage and its determinants among 12-59 months old children in East Africa: a population-based study”

Abstract

- Change the subtitle [introduction] to [background and objectives] or [background and aim]. And try to make it more concise.

We have changed it

- It is recommended to write keywords in alphabetical order.

We have act accordingly. Thank you 

- You may consider changing the keywords to more representative words [Deworming, Soil-transmitted helminth infections, tropical diseases…]

Dear reviewer, thank you for the better professional suggestion. we have added the “tropical diseases” phrase. Thank you 

- The results section is relatively long with extensive statistical results that are clear enough in the results and tables. Try to make it more concise.

Dear reviewer, you are right, the section seems relatively long, however, it was because several significant variables have been found in the final model. Unless, we leave some of them or leaving the confidence interval with the adjusted odds ration we cannot reduce it more. 

Introduction

- Well structured

Thank you

Subjects and methods

- The inclusion/exclusion criteria are not clear. Are there any factors may affect the results.

Dear reviewer we have now put the inclusion and exclusion criteria clearly based on the Guide to DHS statistics. We have referenced it. Furthermore, from the data set of the included countries, children aged 12-59 months who were not assessed for deworming medicine based on the DHS recommendation and had a missing value of the outcome variable were omitted. Similarly, children who had just been died were not included in the study based on the DHS recode manual. Nonetheless, the study includes missing values and "don't know" responses regarding the child's use of deworming medication in the two years before to the interview, but these responses are not regarded as dewormed

Statistical analysis

- Well structured

Thank you.

Discussion

- Well structured

Thank you.

Reviewer #2: Intestinal parasite infections are public health concerns in developing countries which makes the research area is relevant. However, I have some questions and comments on the research work.

Abstract section

1. Avoid using passive sentences. This also applies for the rest of the manuscript sections.

Dear reviewer, thank you. We have act accordingly throughout the manuscript to avoid it.

2. on introduction section line 24, please replace "risk factors" with "associated factors".

We have done it.

3. result section should be written in a manner that attracts the readers attention. e.g. line 41 "having of 3-5, and more children" looks like an incomplete sentence. furthermore, details should be minimized as much as possible to capture the main finding of the study. I highly recommend to rewrite this section

We have act accordingly. Thank you

4. avoid abbreviations from the abstract section if possible. abbreviations like "ANC" should be written in full on its first appearance. line 39. This was also noted on the rest of the manuscript.

Thank you. We have accepted it

Introduction section

It is well organized and provides all the necessary background information.

1. line 103 "literature neglects children under the age of one year, and little is known in East Africa". The rationale for starting deworming among those aged 1 year and above is due to the low likelihood of acquiring the infection even in an endemic setting. The sentence creates a wrong image please remove it.

Thank you. We have removed it 

Method section

1. source and study population: It is not clear how your source and study population are completely different. It needs further elaboration.

Thank you. We have put them clearly now. Our source population included all under five children before the survey period in 12 East African countries, while the study population included children aged 12-59 months in the designated Enumeration Areas (EAs) or primary sample units of the survey clusters.

2. Table 1: the sample size for Ethiopia looks like a typing error. please correct it.

The reviewer is right. We have removed the typos error. Thank you very much

3. Operational definition used by the survey should be available for some of the variables included. e.g traditional contraceptive method, size of the child, wealth index class etc.

We have accepted it. We have prepared an additional table for all variables that we have included in the analysis.

result section

1. line 206-207. it is either 37% increased odds or 1.37 times higher odds not 37% times. In addition, please provide the result next to the category or the variable in consideration to make it more readable.

We have amended it 

2 line 209-210. same as above as well as incomplete sentences and missing figures

3. line 216. "a higher likelihood times"?

We have amended it

4. line 220, 223 missing values

We have amended it

5. line 224 "compared to with mothers".

We have amended it

I wrote those to point out as an example. It is full of grammatical errors and missing figures making it unreadable. please rewrite the result section properly and in an organized manner.

Thank you for the insight. We have improved all the concerns.

---

## [Editor Report · Decision Letter 1]

4 Jan 2024

Deworming coverage and its determinants among 12-59 months old children in East Africa: a population-based study

PONE-D-23-32930R1

Dear Dr. 

We’re pleased to inform you that your manuscript has been judged scientifically suitable for publication and will be formally accepted for publication once it meets all outstanding technical requirements.

Kind regards,

Ayele Mamo Abebe, MSc in pediatric and child health nursing

Academic Editor

PLOS ONE
---

## [Editor Report · Acceptance letter]

22 Jan 2024

PONE-D-23-32930R1 

PLOS ONE

Dear Dr. Terefe, 

I'm pleased to inform you that your manuscript has been deemed suitable for publication in PLOS ONE. Congratulations! Your manuscript is now being handed over to our production team.

Kind regards, 

on behalf of

Assistant professor Ayele Mamo Abebe 

Academic Editor

PLOS ONE